# Bioconversion Study of Olive Tree Biomass Hemicellulosic Hydrolysates by *Candida guilliermondii* at Different Scales for Ethanol and Xylitol Production

Juan Gabriel Puentes [1,2] , Soledad Mateo [1,2,]*, Sebastian Sánchez [1,2] , Inês C. Roberto [3] and Alberto J. Moya [1,2]

1 University Institute of Research on Olive Groves and Olive Oils, GEOLIT Science and Technology Park, University of Jaén, 23620 Mengíbar, Spain; ajmoya@ujaen.es (A.J.M.)
2 Department of Chemical, Environmental and Materials Engineering, University of Jaén, 23071 Jaén, Spain
3 Department of Biotechnology, Engineering College of Lorena, University of São Paulo, Estrada Municipal do Campinho s/n, Cep, Lorena 12602-810, SP, Brazil
* Correspondence: smateo@ujaen.es

**Abstract:** Hemicellulosic biomass from olive-tree pruning (OTPB) was used as a raw material in order to produce a hemicellulosic hydrolysate to be fermented with the non-traditional yeast *Candida guilliermondii* FTI 20037 to obtain ethanol and xylitol. The main objectives of this research were to study the most relevant kinetic parameters involved in the bioconversion process and the correlation between stirred-tank bioreactor and agitated Erlenmeyer flask fermentation. In a first scale-up (using Erlenmeyer flasks) incubated on a rotary shaker at 200 rpm, fermentation assays were performed to determine the most convenient process conditions and the adaptation of the microorganism to the concentrated OTPB and added nutrients culture medium. The best conditions (2.5 kg m$^{-3}$ of initial yeast cells, pH of 5.5 and 30 °C) were set in a bench bioreactor. A comparative study on ethanol and xylitol production was conducted in two scale scenarios, obtaining different results. In the bioreactor, 100% of D-glucose and partially D-xylose were consumed to produce an ethanol yield of 0.28 kg kg$^{-1}$ and an ethanol volumetric productivity of 0.84 kg dm$^{-3}$ h$^{-1}$ as well as a yield and volumetric productivity in xylitol of 0.37 kg kg$^{-1}$ and 0.26 kg dm$^{-3}$ h$^{-1}$, respectively. The kinetic results allowed increasing the action scale and obtaining more real results than the previous steps to enable mini-plant and industrial scaling.

**Keywords:** bioethanol; xylitol; *Candida guilliermondii*; carbohydrates; olive-tree pruning biomass

## 1. Introduction

Lignocellulosic biomass has attracted worldwide interest due to its compositional features (cellulose, hemicellulose and lignin) and for being a widely available renewable and cheap natural resource [1]. Behind cellulose, representing about 15–36% of ligno-cellulosic biomass, hemicellulose is the most common heterogeneous polymer in nature, which is composed especially of pentoses, hexoses and sugar acids [2]. In this sense, it constitutes a high reservoir of carbohydrates that can be used in the fermentation process to obtain value-added metabolites such as xylitol and ethanol [3] but with the handicap of the difficulty of this step. Firstly, it is necessary to control the process of attack of the raw material to obtain the hemicellulosic hydrolysate, usually using acid-based treatment for economic reasons [4,5]. Secondly, a rigorous characterization of the acid hydrolysate obtained after the saccharification process becomes essential because of the release of toxic compounds: acetic acid (present as acetyl groups in the hemicellulose structure), furfural and 5-hydroxymethylfurfural (degradation products of D-xylose and D-glucose, respectively) and aromatic and polyaromatic compounds (degradation components of lignin). These inhibitory compounds, if present in the hydrolysate, should be found only in limited

quantities, as they can affect fermentation to a large extent, depending on the adaptability of the selected microorganism to the medium, also considering the synergistic effect among inhibitory groups [6]. However, the toxic powder of hemicellulose hydrolysates can be palliated by different detoxification technologies: adsorption with activated charcoal, liming, overliming, vacuum evaporation, biological treatment, etc. [7–12].

Olive tree-pruning biomass (OTPB), one of the most important crops in Spain, other Mediterranean areas (98% of total cultivated olive trees), American countries, Australia, India and China, is a great agro-based lignocellulosic biomass source. This kind of feedstock originated due to the need for optimizing the production of olive fruit and as a consequence of the tree rejuvenation by pruning. This operation entails the production of $1–10$ t ha$^{-1}$ of OTPB for super-high density olive groves, while high-density orchards produce $1.2–18.5$ t ha$^{-1}$ [13,14] and are experiencing continuous expansion around several countries linked to Mediterranean-like climates, reaching, in 2020, 12.8 Mha of the world's olive-growing surface [15]. Approximately 40% of this area is in the European Union (5.15 Mha), with Spain occupying the first place with 2.6 Mha, since it is considered the largest producer of olive oil in the world. The major characteristic of OTPB is its high compositional heterogeneity, being mainly composed of D-xylose, D-glucose and other sugars (L-arabinose, D-galactose and D-mannose). As inhibitors, acetic acid, phenolic compounds, levulinic and formic acids, generated during the saccharification process, and furans from sugar degradation [16] can be produced. The large volume of biomass, together with the serious environmental damage caused by its uncontrolled burning on the farm lands, has suggested the possibility of exploiting different lignocellulosic raw materials to produce oligosaccharides and monomeric carbohydrates [3] to be fermented for the bioproducts production [14].

Xylitol, a natural C5 sugar alcohol with sweetness similar to sucrose, is used in the pharmaceutical and food industry and has been obtained since the 1970s by the catalytic hydrogenation of D-xylose, although involving various purification steps, so increasing the final cost and hindering its market entry compared to sorbitol or sucrose. Therefore, it is necessary to develop and optimize the biotechnological production of xylitol in order to improve the market competitiveness [17]. In this way, the oxygen transfer rate ($K_{La}$) appears as a quite important parameter for fermentative microorganisms if the objective is a scale change [18].

Ethanol, a renewable biofuel obtained by sugar fermentation, has been reported as a partial replacement of fossil fuels [19]. Nevertheless, nowadays, there is an increasing demand of food substrates such as sugarcane or maize starch since they are an industrial supplier, therefore causing a double conflict related to both social and environmental drawbacks derived from high pressure on food supplies and deforestation due to plantations expansion [20]. In this way, it is relevant to focus the production of fuels and high value-added products on the use of raw materials that do not compete with food sources, so lignocellulosic biomass could be an excellent alternative.

The biotechnological approach based on the utilization of lignocellulosic biomass is strongly dependent on the yeast used in the process. In this sense, *Candida guilliermondii*, an ascomycetes yeast widely distributed in the nature, has attracted high interest in both biotechnological [21–23] and industry research because of its capacity to metabolize C5 and C6 sugars to value-added metabolites, such as xylitol and ethanol [24]. The control of some determinant variables such as initial microorganism concentration [25], pH [26], temperature [27], initial sugar concentration [6], toxic compounds concentration [28] and aeration level are of particular interest for the fermentation process. This last variable has a determinant effect on the xylitol production, because it is directly related to the imbalance between biomass or xylitol accumulation, which is an effect controlled by D-xylose reductase and xylitol dehydrogenase [29].

At the laboratory scale, it is essential to know the interaction between relevant independent variables in fermentation processes and the controlled output parameters. Experimental trials are usually conducted, in the first instance, using simple reactor configurations

of small size in order to establish an appropriate value for the most influential variables in fermentation. However, the results of the research could vary when experiments are conducted under similar conditions but the equipment capacity is modified as well as, in some cases, the configuration of the system to allow operating with a larger volume of fermentation medium. In this sense, it is very interesting to evaluate the similarity of the experimental results obtained with the scale-up of the fermentation system toward configurations more in line with the industrial level. Therefore, the main purpose of this work is to study the fermentative behavior of *Candida guilliermondii* FTI 20037 comparing both Erlenmeyer flasks and bench bioreactor operation systems using OTPB hemicellulosic hydrolysate, with high simple sugars concentration, obtained by dilute $H_2SO_4$ hydrolysis, thus controlling the D-glucose and D-xylose uptake and ethanol and xylitol production in order to conclude if it is convenient to use OTPB hemicellulosic hydrolysate as a biotechnological substrate to produce these bioproducts of commercial interest.

## 2. Materials and Methods

### 2.1. Raw Material

OTPB (*Olea europaea*) was collected on a farm land in the Andalusia region (within 411,730–411,740 m EW and 4,196,882–4,196,893 m NS relative to UTM coordinates). After pruning operation on twenty-year olive trees, representative and random fresh branches located 1.5 m above the ground were deprived of their leaves and milled "in situ" by a chopper (Halcon PH320I). At that moment, the sample was collected in 50 $cm^3$ plastic holed boxes and transferred to the research laboratory, where all material was expanded into a ventilated room and air-dried at room temperature until reaching equilibrium moisture content (w $\simeq$ 8%) and subsequently milled using a laboratory hammer mill (Retsch GMBH mod. SM11) and fraction graded with a plates sifter (Retsch Vibro) to a particle size in range of 30–40 mesh number, according to ASTM E 11–87 guidelines. Finally, this raw material was stored in glass jars sealed until use.

In addition, one reason to eliminate leaves from this biomass is based on the usefulness of this olive tree-derived material separately, so promoting the possibility of concentrating a new interesting by-product of this biotechnological process that could increase the economic benefit [24]. Moreover, in general, the high percentage of this fraction in OTPB (25%) along with its high content of lignin (40% wt. dry basis, [30]) could hinder the bioethanol generation.

### 2.2. General Analytical Methods

The quantification of carbohydrates (D-glucose and D-xylose), acetic acid, xylitol and ethanol concentrations was conducted by high-performance liquid chromatography (HPLC) using a Waters chromatograph equipped with a Bio-Rad Aminex HPX-87H (300 $\times$ 7.8 mm) column and a refractive index detector. Before injection, the samples were diluted and filtered through a Waters Sep-Pak C18 filter. The operation conditions were: 0.49 kg m$^{-3}$ of $H_2SO_4$ as mobile phase at a flow rate of 0.6 $cm^3$ min$^{-1}$, 45 °C column temperature and injection volume of 0.02 $cm^3$.

NREL methodology was used for extractives (nonstructural components such as pectins, fats, terpenes, phenols, tannins, uronic acids, etc.) determination. Moisture was determined by constant dry weight according to TAPPI norm T11 m-59 and ash by the difference in weight before and after incineration at 575 $\pm$ 25 °C by the procedure proposed in the TAPPI norm T15 os-58. To calculate the cellulose, hemicellulose and Klason lignin percentages, the methodology proposed by Irick et al. [31] was used, whereas the method proposed by Rocha [32] was developed to determine acid-soluble lignin.

Furfural and hydroxymethylfurfural (HMF) were also determined by Waters HPLC but with an UV absorbance detector (at 276 nm) and a Waters Resolve 5 μm Spherical C18 (300 $\times$ 3.9 mm) column at room temperature, using an acetonitrile/water ratio of 1/8 (with an acetic acid mass fraction of 1%) as the mobile phase at 0.8 $cm^3$ min$^{-1}$ and sample

volume of 0.020 cm$^3$. Before injection, diluted samples were filtered through Millipore membranes HAWP04700 with 0.45 μm pore size.

The cell concentration, $x$, was determined using a BECKMAN DU 640 model spectrophotometer, at 600 nm, by means of a calibration curve (dry weight vs. optical density) obtained from cells grown on hydrolysate or synthetic medium agitated on a rotary shaker at 200 rpm, 30 °C, for 72 h, as shown in Equation (1).

$$x = 0.235 A_{600} - 0.0001 \qquad (1)$$

The total phenolic compounds content, $TFC$, expressed as concentration of ferulic acid, was determined colorimetrically using a BECKMAN DU 640 model spectrophotometer at 760 nm, applying the Folin–Ciocalteu modified method, as shown in Equation (2) [33].

$$TFC = 20.63 A_{760} - 0.4824 \qquad (2)$$

The volumetric oxygen transfer coefficient ($K_{La}$) was determined under a fixed flow rate (0.4 vvm) and stirring rate (350 rpm) on the fermentation media by an indirect method (without cell) using an ingold polarographic electrode. The procedure proposed by Silva et al. [34] and the mathematical equation proposed by Stanbury et al. [35] were used.

### 2.3. Acid Hydrolysis

Acid hydrolysis of olive-tree pruning biomass was conducted in a 50 dm$^3$ stainless steel discontinuous reactor and stirred tank provided with a system of electrical resistance heating and agitation by rotation about its own axis, using a 2.5% $H_2SO_4$, 1:10 solid/liquid ratio ($w/v$) for 150 min at 100 °C. These acid hydrolysis conditions were chosen from a previous work [16] with the aim of obtaining the highest concentration of total sugars in the hydrolysate. The solid and liquid phases were separated by filtration. The solid phase was washed until neutrality and stored for subsequent analysis. The hydrolysate was concentrated under vacuum at 65 ± 5 °C and stored at 4 °C.

### 2.4. Microorganism and Inoculum Cultivation

*Candida guilliermondii* FTI 20037 was stored at 20 °C in 100 cm$^3$ test tubes on a sterilized solid culture medium composed of (in kg m$^{-3}$): yeast extract 3; malt extract 3; peptone 5; D-xylose 10; agar 20. For inoculum preparation, fresh yeast cells obtained after incubation at 30 °C, for 24 h, were transferred into 15 cm$^3$ test tubes containing 5 cm$^3$ of sterile distilled water. Aliquots of 1 cm$^3$ of this cellular suspension were introduced into 125 cm$^3$ Erlenmeyer flasks containing 50 cm$^3$ of medium described above. The resulting suspension was stored at 4 °C until use. The cultures were incubated at 30 °C on a rotary shaker at 20.94 rad s$^{-1}$ for 30 h. Cells were separated by centrifugation (1100× $g$ for 20 min) and then used to inoculate the fermentation medium.

### 2.5. Fermentation Conditions

Fermentation assays in Erlenmeyer flasks were conducted in 125 cm$^3$ flasks containing different amounts of medium (25–50–75 cm$^3$), representing values of $\omega$ ranging from 0.2 to 0.6. The culture medium was supplemented by the addition of nutrients proposed either by Lindegren et al. [36], including some modifications thereof, or by Roberto et al. [37].

The first series of experimental trials, in Erlenmeyer flasks, was performed considering the following values of the variables: 200 rpm; pH = 5.0; $\omega$ = 0.4 and 1.5 kg m$^{-3}$ inoculum concentration. The most suitable type of medium was studied to carry out the subsequent fermentative tests. The flasks were closed with hydrophobic cotton. The pH of the medium was adjusted using either HCl or NaOH solutions.

A new experimental series was also performed, in Erlenmeyer flasks, once the amounts of nutrients to be supplemented were selected to analyze the effect of variables such as temperature (25–30–35 °C), initial inoculum concentration (0.5–1.5–2.5 kg m$^{-3}$) and aeration (0.2–0.4–0.6), keeping fixed both pH and stirring speed at 5.0 and 200 rpm, respectively.

The third experimental series, using Erlenmeyer flasks, aims to elucidate whether the pH = 5, set in the previous experimental series, is the most appropriate for fermentation trials in bioreactors, analyzing different pH values: 3.0; 4.0; 5.0; 5.5; 6.0 and considering the variables inoculum concentration, temperature, aeration and agitation established from previous series.

Other tests were carried out in Erlenmeyer flasks to contract the improvements that could result from an increase in the initial inoculum concentration with respect to that established in previous series (1.5 kg m$^{-3}$), using two different biomass concentrations: 2.5 kg m$^{-3}$ and 3.5 kg m$^{-3}$.

After fermentation in Erlenmeyer flasks, some tests were carried out on a 2 dm$^3$ stirred-tank bioreactor model Biostat B (B. Braun Biotech International) equipped with controllers of temperature, agitation, aeration and pH; one turbine with six-plane blades was used. The best conditions obtained using Erlenmeyer flasks were applied to bioreactor assays. The aeration rate was 0.4 vvm, the agitation speed was 350 rpm and the volume of the medium was set at 1 dm$^3$. The process was monitored by periodic sampling to determine cell growth, D-glucose and D-xylose consumption and ethanol and xylitol production.

### 2.6. Fermentative Parameters Calculation

Xylitol global yield factor ($Y_{Xy/s}^G$, kg kg$^{-1}$) was defined from the adjustment of xylitol concentration ($Xy$) vs. consumed D-xylose concentration. Ethanol global yield factor ($Y_{E/s}^G$, kg kg$^{-1}$) was defined from linear graphic representations of ethanol concentration ($E$) vs. consumed D-glucose plus D-xylose concentrations. Cell global yield factor ($Y_{x/s}^G$, kg kg$^{-1}$) was defined from the linear adjustment of total cells concentration and the total amount of substrate (D-xylose and D-glucose) consumed. Xylitol volumetric productivity ($Q_{Xy}$, kg m$^{-3}$ h$^{-1}$) was calculated as the ratio between the concentration of xylitol per time unit. Ethanol volumetric productivity ($Q_E$, kg m$^{-3}$ h$^{-1}$) was calculated as the ratio between the concentration of ethanol per time unit. Cell volumetric productivity ($b$, kg m$^{-3}$ h$^{-1}$) was determined by quadratic fit as described by Sánchez et al. [38]. The specific rate of total substrate consumption ($q_s$, kg kg$^{-1}$ h$^{-1}$), sum of D-glucose and D-xylose, has been calculated by the differential method of treatment of kinetic data.

## 3. Results and Discussion

### 3.1. Raw Material Composition

According to Puentes et al. [16], the original OTPB shows some general features with respect to other widespread industrially agro-crops and woody species. In the biomass batch used in this work, the hemicellulose represented 18.63 ± 0.27% on a dry basis (Table 1), which is a lower content than the values shown for sugarcane bagasse (25–30%) [39–41], wheat straw (23–26%) [42], rice straw (35%), poplar (30%), eucalyptus (30%), giant reed (22%) [10,12], pine (35%), or sorghum bagasse (36%) [43] but with a high percentage of the major component, cellulose, 33.85 ± 0.76%, which is in this case similar to other lignocellulosic biomasses such as sugarcane bagasse (32–44%), [44–46] or wheat straw (29–35%) [47,48], which are widely used to obtain biofuels by means of hydrolytic and fermentative processes. Therefore, considering the potential recovery of monomeric sugars from both hemicellulose and cellulose fractions, OTPB is postulated as a suitable source of fermentable sugars for ethanol and xylitol production. The total lignin (23.12 ± 0.04% of dry weigh) expressed as the sum of insoluble (18.93 ± 0.08%) and soluble lignin (4.19 ± 0.03%) showed similar values in comparison to sugarcane bagasse (19–24%) [44,49] or eucalyptus (20–27%) [50,51]. In this regard, other authors have published the total lignin composition for OTPB in the range 9–24% [52–56], so the current OTPB components are in the range above mentioned, although the noted divergences could be explained not only by differences based on the geographical origin of this type of biomass but also by the compositional variation of the raw material, depending on the amount, for example, of leaves whose lignin content seems to be high, accordingly to some authors (40% [57], 26% [58]).

**Table 1.** OTPB characterization, mass fraction as % of dry weight.

| Fraction | % |
|---|---|
| Cellulose | $33.85 \pm 0.76$ |
| Hemicellulose | $18.63 \pm 0.27$ |
| Acid-insoluble lignin | $18.93 \pm 0.08$ |
| Acid-soluble lignin | $4.19 \pm 0.03$ |
| Ash | $4.62 \pm 0.28$ |
| Acetyl groups | $2.07 \pm 0.02$ |
| Water extracts | $17.39 \pm 0.32$ |
| Ethanol extracts | $1.79 \pm 0.01$ |

Average data of at least three determinations.

Ash content ($4.62 \pm 0.28$%), directly related to lignin percentage [57], represents a similar value to other feedstocks (sugarcane bagasse [39]), but it is higher than other batches of OTPB characterized in the bibliography (0.5–2.3%) [24,52]. Regarding the extractives fraction ($19.18 \pm 0.16$%), which is generally encompassing organic acids, inorganic substances, chlorophyll, nitrogenous material and non-structural carbohydrates [57], it was recovered using a sequential water and ethanol extraction. The fact that the amount of aqueous and ethanolic extracts from this feedstock showed a value close to 19%, which is quite similar to the extracts contained in olive tree pruning consisting exclusively of wood (almost 28%) but much lower than those ones in olive tree leaves (37–45% [54,57,58]) or complete olive tree pruning, including branches, leaves and wood (36–37% [57]), could corroborate the absence of leaves in the starting material. For this reason, it could be expected to be a high accessibility and reactivity powder when the leaves are removed during the chemical attack. Finally, the present OTPB batch structural components were consistent with those based on literature data, cellulose (22–37%), hemicellulose (14–21%) and lignin (9–28%) [16,24,52,55–62], although it is necessary to explain that the characterization results are highly influenced by the methodology applied due to the particular OTPB structure where, besides cellulose, D-glucose can be found in the parenchymal zone, as polyphenolic glycoside [9], which should be considered as part of hemicellulose and not as cellulose. For this reason, care must be taken when a method based on monosaccharides to explain cellulose and hemicellulose content is used [63]. Furthermore, the OTPB composition clearly varies among research works due to its structural heterogeneity, as this material is composed of leaves, branches and wood, and the percentages of these fractions with respect to the total biomass may be different depending on the sampling area and type of pruning, even differing although the material was obtained from the same crop [64].

### 3.2. OTPB Hemicellulosic Hydrolysate

Results for both original and concentrated hydrolysates from acid treatment are shown in Figure 1, summarizing data of D-xylose, D-glucose and L-arabinose concentrations as well as the main inhibitors (acetic acid, total phenols and furans). After the evaporation process, the recoveries of simple sugars increased 2.7 times (concentration factor), approximately, and practically no sugar losses were detected, although this did not occur with the concentrations of inhibitory compounds. Thus, the use of vacuum evaporation as a detoxification stage is noteworthy, since it decreases the concentration of toxic compounds for yeasts in the hydrolysate, which was also noted by some authors [9,65].

The sugar composition of OTPB hemicellulosic hydrolysate exhibited a higher D-xylose content than D-glucose content, which had already been shown in previous research for this same type of raw material, which was devoid of leaves [62]. Similar concentrations of D-glucose and D-xylose have been obtained by subjecting OTPB to acid hydrolysis treatments (with oxalic, sulfuric and phosphoric acids) in other investigations [59,60,62,66], although it must be taken into account that apart from variables of interest such as the concentration of hydrolytic agent, temperature and time, the concentration of sugars in the hydrolysate is strongly dependent on the type of treatment and the solid: liquid ratio employed.

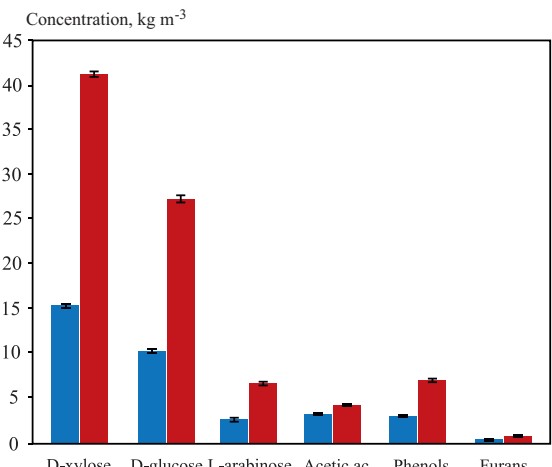

**Figure 1.** Sugars and inhibitors concentrations in kg m$^{-3}$. Raw hydrolysate profile before (blue column) and after concentration (red column).

Acetic acid, recognized as a compound with dual inhibitory and substrate effect [28,67] and the main aliphatic acid present in OTPB hydrolysates [9], was removed until 50%, so its final concentration in the concentrated OTPB hydrolysate was 3.99 kg m$^{-3}$. The very low pH of the OTPB hydrolysate (pH $\simeq$ 1) could have favored the partial elimination of this compound by volatilization, since it would be basically in its undissociated form [68]. Anyway, the presence of this compound has a double substrate/inhibitory effect, and although it can slow down the fermentation process [69], *C. guilliermondii* could consume up to 62% of acetic acid from a starting concentration of 8.1 kg m$^{-3}$ [70].

On the other hand, reductions of both phenolic compounds (lignin degradation products responsible for the membrane integrity lost) [24] and total furans (from sugars decomposition) were 12.8% and 27.1%, respectively. The decrease of this last group of inhibitory compounds concentrations was evinced considering that the furfural boiling point under vacuum evaporation is 50–55 °C [71], which is the temperature reached under the experimental conditions used in the concentration stage. In addition, although the phenolic compounds would reduce the rate of fermentation, *C. guilliermondii* would have the capacity to reduce the toxic effect of these compounds, since it can convert phenolics into their corresponding acids or alcohols [72]. Regarding furans, negative effects on the cellular development of this yeast have been reported for concentrations of 0.67 kg m$^{-3}$ [69]. In any case, the concentrations of acetic acid (3.99 kg m$^{-3}$) as well as phenolic and furanic compounds (6.94 kg m$^{-3}$ and 0.55 kg m$^{-3}$, respectively) were found in the concentrated hydrolysate in quantities not high enough to be considered excessively toxic to *C. guilliermondii* yeast. In this regard, other research works, in which OTPB hemicelullosic hydrolysate was used under similar experimental conditions to this study (H$_2$SO$_4$ 2%, 120 °C and 90 min), reported acetic acid concentrations of 1.7 kg m$^{-3}$ or 2.2 kg m$^{-3}$ [9,42]. However, some authors established the toxicity level of acetic acid for *C. guilliermondii* above 1 kg m$^3$ and a very strong inhibitory character for concentrations higher than 3 kg m$^{-3}$ [71], while in the case of furfural, a limit of 1 kg m$^{-3}$ was reported [73].

After hydrolytic treatment, the combined severity factor, log $R_0$, calculated by Equation (3), was 3.11. In this equation, *T* represents the temperature in Celsius degrees and *t* represents the time.

$$log\ R_0 = log \int_0^t e^{\frac{T-100}{14.75}}\,dt - pH \tag{3}$$

This value of log $R_0$ represents low-medium severity conditions [52,74,75], seeking a limited or no simple sugars degradation from the hydrolysate. In any case, the presence of inhibitory compounds in the concentrated hydrolysate could implied the need for minimizing the amount of toxic compounds for some microorganisms, which is the objective of various studies to obtain a suitable culture medium, promoting yeast growth and im-

proving the bioproducts production [76,77]. Fortunately, it is not the case for the current study, as a detoxification step was not required, which is an important aspect in order to achieve the economy of the process by reducing subsequent costs [78]. It should be added that the use of a high concentration of substrate minimizes the size of the fermentation reactors and therefore their cost for a given product capacity, and it even increases the final concentration of bioproducts, also encouraging the economic viability of the final separation process [17]. In this sense, the cellulose-rich solid residue that remains after the acid hydrolysis of the hemicellulosic fraction can and should be valorized. For this purpose, enzymatic hydrolysis and subsequent fermentation with traditional yeasts would not pose major problems. On the other hand, it can also be used to obtain nanocellulose, whose industrial applications are very promising [79]

### 3.3. Fermentation of OTPB Concentrated Hemicellulosic Hydrolysate at Different Scales

Both ethanol and xylitol production from OTPB concentrated hemicellulosic hydrolysate were previously tested. Firstly, different nutritional mediums with initial monomeric sugars (mainly D-glucose and D-xylose) composition close to $23 \pm 2\,\mathrm{kg\,m^{-3}}$ and $36 \pm 3\,\mathrm{kg\,m^{-3}}$, respectively, were used. Subsequently, tests to determine the main variables that affect the fermentation process, such as the initial concentration of yeast cells, temperature, aeration ($\omega$) and pH, were performed. Therefore, in order to study the influence of the substrate type on the fermentation process of *C. guilliermondii* yeast, trials with both OTPB hydrolysate and synthetic medium were carried out. Preliminary experiments based on acid hydrolysates involved the study of three alternatives: OTPB concentrated hydrolysate without the addition of nutrients (OTPB-H), OTPB concentrated hydrolysate with the addition of nutrients proposed by Roberto et al. [37] (OTPB-RH) and with the addition of nutrients proposed by Lindegren et al. [36] at 50% (OTPB-LH). On the other hand, with respect to the experimental tests using a synthetic medium, two possibilities were explored: substrate constituted by the addition of nutrients proposed by [37] (SM-R) or according to [36] at 50% (SM-L). For these experiments, the initial inoculum concentration was fixed at $1.5\,\mathrm{kg\,m^{-3}}$, temperature $30\,°C$, pH = 5 and $\omega = 0.4$. *C. guilliermondii* was unable to ferment OTPB concentrated hydrolysate without the addition of nutrients (OTPB-H). On the contrary, it metabolized the supplemented hydrolysates (OTPB-RH and OTPB-LH), thereby resulting in the growth of yeast cells, substrate consumption and bioproducts generation, as shown in Table 2.

**Table 2.** Fermentative results obtained in preliminary assays using different culture media with fixed fermentative parameters: $x_0 = 1.5\,\mathrm{kg\,m^{-3}}$, $T = 30\,°C$, pH = 5 and $\omega = 0.4$.

| Parameter | OTPB-RH | OTPB-LH | SM-R | SM-L |
|---|---|---|---|---|
| $Y_{x/s}^{G}$, kg kg$^{-1}$ | $0.05 \pm 0.01$ | $0.06 \pm 0.01$ | $0.10 \pm 0.01$ | $0.09 \pm 0.01$ |
| $Y_{E/s}^{G}$, kg kg$^{-1}$ | $0.33 \pm 0.01$ | $0.36 \pm 0.01$ | $0.19 \pm 0.01$ | $0.38 \pm 0.01$ |
| $E_{max}$, kg m$^{-3}$ | $15.09 \pm 0.01$ [88] | $18.42 \pm 0.18$ [88] | $6.16 \pm 0.03$ [40] | $15.15 \pm 0.05$ [40] |
| $Q_E$, kg m$^{-3}$ h$^{-1}$ | $0.17 \pm 0.01$ [64] | $0.24 \pm 0.01$ [64] | $0.32 \pm 0.01$ [16] | $0.57 \pm 0.01$ [16] |
| $Y_{Xy/s}^{G}$, kg kg$^{-1}$ | $0.25 \pm 0.01$ | $0.39 \pm 0.01$ | $0.56 \pm 0.01$ | $0.40 \pm 0.01$ |
| $Xy_{max}$, kg m$^{-3}$ | $7.98 \pm 0.01$ [112] | $10.84 \pm 0.17$ [88] | $24.99 \pm 0.31$ [64] | $13.74 \pm 0.02$ [88] |
| $Q_{Xy}$, kg m$^{-3}$ h$^{-1}$ | $0.07 \pm 0.01$ [64] | $0.12 \pm 0.01$ [88] | $0.54 \pm 0.01$ [40] | $0.19 \pm 0.01$ [64] |

Superscripts indicate time (h). $r^2 > 0.953$ for all linear fits. SM (synthetic medium). H (fermentation using concentrated hydrolysate), R (nutrients supplement proposed by [37]), L (nutrients supplement proposed by [36] at 50%).

In general, for synthetic medium, similar biomass yields for both SM-R and SM-L media were obtained, but there were higher ethanol parameters (production, yields and productivities) for SM-L. However, SM-R was more effective for xylitol production. On the other hand, comparing the results for supplemented hydrolysates (L or R media), better results were obtained for OTPB-LH fermentation. Consequently, it was evinced that OTPB-LH and SM-L, as shown in Figure 2, showed the best fermentative results with parameters

values quite close to each other although slightly superior for synthetic media, with global yields in ethanol and xylitol of 0.36 and 0.39 kg $kg^{-1}$, respectively, in the case of the OTPB-LH and 0.38 and 0.40 kg $kg^{-1}$, respectively, in the SM-L assay. Likewise, it has been also evinced that this yeast was able to rapidly consume D-glucose in SM-L (Figure 2A), taking only 14 h, while for OTPB hydrolysate supplemented with these same nutrients (OTPB-LH), the time for total consumption of this hexose was longer (42 h) (Figure 2B). On the other hand, for both hydrolysate and synthetic "L" culture media, D-xylose was consumed after 112 h, although the consumption rate of the pentose was higher when using SM-L. Furthermore, *C. guilliermondii* exhibited, in both synthetic and hydrolyzed media, a non-diauxic behavior with the simultaneous consumption of D-glucose and D-xylose. Reasons for choosing OTPB hemicellulosic hydrolysate as a culture medium are mainly based on the price and availability of the raw material to produce a suitable carbon source which would reduce process costs [17]. Consequently, OTPB-LH has been used as substrate fermentation by *C. guilliermondii*, so reaching an ethanol yield in consonance with the results obtained by Mateo et al. [24]. However, the global xylitol yields reported by these authors were lower than those achieved in this research, which are much more similar to those obtained by Rao et al. [80]. These results confirm that the addition of nutrients proposed by Lindegren et al. [36] at 50% is the most reliable decision to ferment hydrolysates from olive tree pruning or other olive tree grove by-products [53,66,74,81,82].

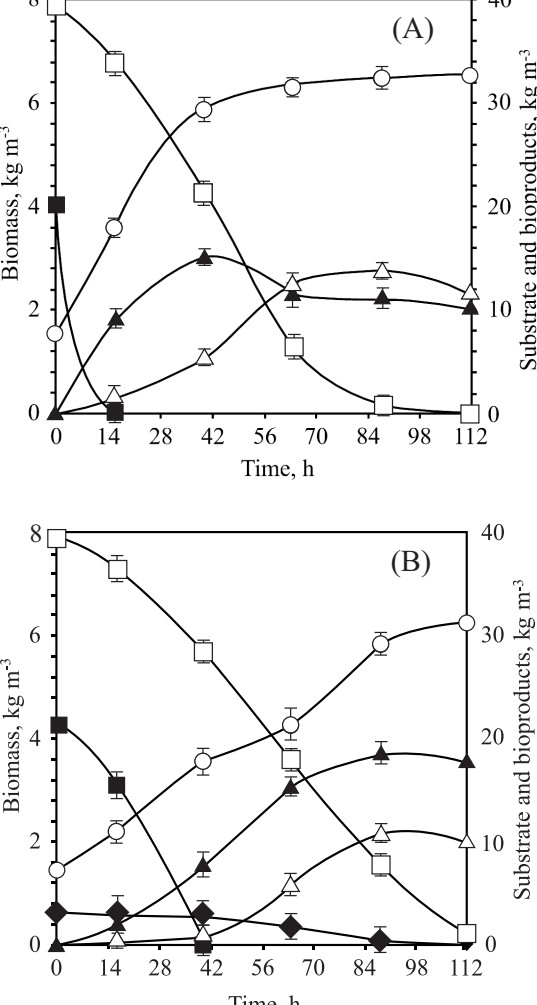

**Figure 2.** Fermentation profiles of experiments in Erlenmeyer flasks by *C. guilliermondii*. (**A**) SM-L. (**B**) OTPB-LH. Biomass (hollow circle), D-xylose (hollow square), D-glucose (filled square), ethanol (filled triangle), xylitol (hollow triangle) and acetic acid (filled rhomb).

Subsequently, and taking into account the most favorable conditions with regard to the supplemented nutritional medium for OTPB hydrolysates, a new series of experiments was designed to test the effect of different fermentative parameters such as temperature (25–30–35 °C), initial inoculum concentration (0.5–1.5–2.5 kg m$^{-3}$) and the $\omega$ value (0.2–0.4–0.6), while the pH (5.0) and agitation (200 rpm) were fixed. It is worth noting that *C. guilliermondii* was not able to ferment hydrolysates at 35 °C for any of the tested pH values and cell concentrations.

The effect of biomass concentration in the medium at 25 °C can be observed in (Table 3) so noting, in general, a maintenance or slight improvement (in ethanol and xylitol yields) of fermentative parameters when the inoculum concentration rose from 0.5 kg m$^{-3}$ to 2.5 kg m$^{-3}$ for low or high aeration levels, considering OTPB-L (1 and 2) or OTB-L (3 and 4), respectively. Regarding the initial inoculum concentration, several authors have reported the need to start the fermentation process using a high concentration of cells [25,83], but other researchers [84] observed a negative effect on xylitol production by *C. guilliermondii* when increasing the inoculum levels from 1 to 4 kg m$^{-3}$ of cells in sugarcane bagasse hemicellulosic hydrolysate. Conversely, an increase of the aeration level (from $\omega = 0.2$ to $\omega = 0.6$), comparing OTPB-L (1 and 3) or OTPB-L (2 and 4) assays, correspondingly, seemed to favor ethanol but especially xylitol yield. The data collected in Table 3 show that the most favorable achieved results were those obtained considering intermediate parameter values ($x_0 = 1.5$ kg m$^{-3}$, T = 30 °C and $\omega = 0.4$, OTPB-L5). Under these positive conditions, global yields ($Y_{E/s}^G = 0.36$ kg kg$^{-1}$ and $Y_{Xy/s}^G = 0.38$ kg kg$^{-1}$), volumetric productivities ($Q_E = 0.14$ kg m$^{-3}$ h$^{-1}$ and $Q_{Xy} = 0.07$ kg m$^{-3}$ h$^{-1}$) and target concentrations in by-products ($E_{max} = 12.94$ kg m$^{-3}$ and $Xy_{max} = 9.15$ kg m$^{-3}$) were the highest results, in consonance to [85,86].

**Table 3.** *C. guilliermondii* fermentations series designed for temperature (25–30–35 °C), initial inoculum concentration (0.5–1.5–2.5 kg m$^{-3}$) and the $\omega$ value (0.2–0.4–0.6), with fixed pH (5.0) and agitation (200 rpm), and adding nutrients supplement proposed by [36] at 50%.

| Experiment | OTPB-L1 | OTPB-L2 | OTPB-L3 | OTPB-L4 | OTPB-L5 |
|---|---|---|---|---|---|
| $Y_{x/s}^G$, kg kg$^{-1}$ | 0.11 ± 0.01 | 0.10 ± 0.01 | 0.08 ± 0.01 | 0.07 ± 0.01 | 0.07 ± 0.01 |
| $Y_{x/s+Ac}^G$, kg kg$^{-1}$ | 0.10 ± 0.01 | 0.10 ± 0.01 | 0.08 ± 0.01 | 0.07 ± 0.01 | 0.07 ± 0.01 |
| $Y_{E/s}^G$, kg kg$^{-1}$ | 0.11 ± 0.01 | 0.25 ± 0.01 | 0.27 ± 0.01 | 0.33 ± 0.01 | 0.36 ± 0.01 |
| $E_{max}$, kg m$^{-3}$ | 8.68 ± 0.31 [144] | 8.08 ± 0.06 [120] | 10.19 ± 0.33 [144] | 10.17 ± 0.74 [120] | 12.94 ± 0.07 [96] |
| $Q_E$, kg m$^{-3}$ h$^{-1}$ | 0.06 ± 0.01 [144] | 0.07 ± 0.01 [120] | 0.06 ± 0.01 [168] | 0.10 ± 0.01 [96] | 0.14 ± 0.01 [72] |
| $Y_{Xy/s}^G$, kg kg$^{-1}$ | 0.23 ± 0.01 | 0.26 ± 0.01 | 0.35 ± 0.01 | 0.36 ± 0.02 | 0.38 ± 0.02 |
| $Xy_{max}$, kg m$^{-3}$ | 7.38 ± 0.08 [168] | 7.88 ± 0.13 [192] | 7.68 ± 0.33 [192] | 8.15 ± 0.05 [168] | 9.15 ± 0.34 [144] |
| $Q_{Xy}$, kg m$^{-3}$ h$^{-1}$ | 0.04 ± 0.01 [168] | 0.05 ± 0.01 [192] | 0.04 ± 0.01 [192] | 0.06 ± 0.01 [144] | 0.07 ± 0.01 [120] |

Superscripts indicate time (h). r$^2$ > 0.958 for all linear fits. OTPB-L1 ($x_0 = 0.5$ kg m$^{-3}$, T = 25 °C and $\omega = 0.2$), OTPB-L2 ($x_0 = 2.5$ kg m$^{-3}$, T = 25 °C and $\omega = 0.2$), OTPB-L3 ($x_0 = 0.5$ kg m$^{-3}$, T = 25 °C and $\omega = 0.6$), OTPB-L4 ($x_0 = 2.5$ kg m$^{-3}$, T = 25 °C and $\omega = 0.6$) and OTPB-L5 ($x_0 = 1.5$ kg m$^{-3}$, T = 30 °C and $\omega = 0.4$).

In line with the most favorable conditions described up to now ($x_0 = 1.5$ kg m$^{-3}$, pH = 5.0, T = 30 °C, $\omega = 0.4$ and agitation = 200 rpm), the third experimental series of fermentation experiments was designed to understand the effect of the initial pH, considering different values (3.0, 4.0, 5.0, 5.5, and 6.0) for this parameter as the pH range used in other fermentation studies with *C. guilliermondii* was quite variable, so this aspect is not clear enough [28,70]. It is worth noting that for pH values below 5.0, the fermentation processes did not progress favorably. Therefore, according to Figure 3A, the best global yields of both ethanol ($Y_{E/s}^G = 0.35$ kg kg$^{-1}$) and xylitol ($Y_{Xy/s}^G = 0.44$ kg kg$^{-1}$) were reached at pH = 5.5. It should be noted that for this pH value, significant ethanol and xylitol productivities ($Q_E = 0.53$ kg m$^{-3}$ h$^{-1}$ and $Q_{Xy} = 0.17$ kg m$^{-3}$ h$^{-1}$) as well as maximum concentrations of these bioproducts ($E_{max} = 13.88$ kg m$^{-3}$ and $Xy_{max} = 8.21$ kg m$^{-3}$) were obtained.

Due to the controversy among authors about the initial concentration of the inoculum, the fermentation study using Erlenmeyer flasks was finally carried out at the pH previously optimized (5.5), varying the initial concentration of the inoculum (1.5, 2.5 and 3.5 kg m$^{-3}$) and fixing T = 30 °C and $\omega$ = 0.4 (Figure 3B). In this case, the use of the intermediate inoculum concentration (2.5 kg m$^{-3}$) resulted in the best values for global ethanol and xylitol yields, 0.38 and 0.46 kg kg$^{-1}$, respectively. These conditions involved volumetric productivities of 0.53 kg m$^{-3}$ h$^{-1}$ for ethanol and 0.17 kg m$^{-3}$ h$^{-1}$ for xylitol, as well as maximum ethanol and xylitol concentrations of 13.88 kg m$^{-3}$ and 8.21 kg m$^{-3}$, correspondingly (Figure 4). Under optimal inoculum conditions, D-glucose was consumed in only 24 h, while all D-xylose was assimilated in 120 h.

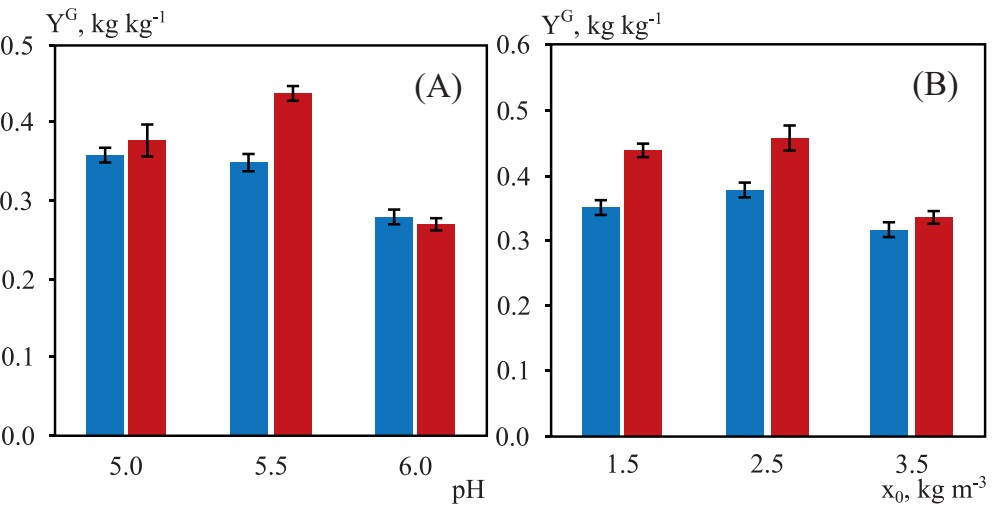

**Figure 3.** Ethanol yields, $Y_{E/s}^{G}$, (blue column) and xylitol yields, $Y_{Xy/s}^{G}$, (red column) in the fermentation by *C. guilliermondii* using OTBP with nutrients proposed by [36] at 50%. (**A**) $x_0$ = 1.5 kg m$^{-3}$, T = 30 °C, $\omega$ = 0.4, agitation speed 200 rpm. (**B**) pH = 5.5, T = 30 °C, $\omega$ = 0.4, agitation speed 200 rpm.

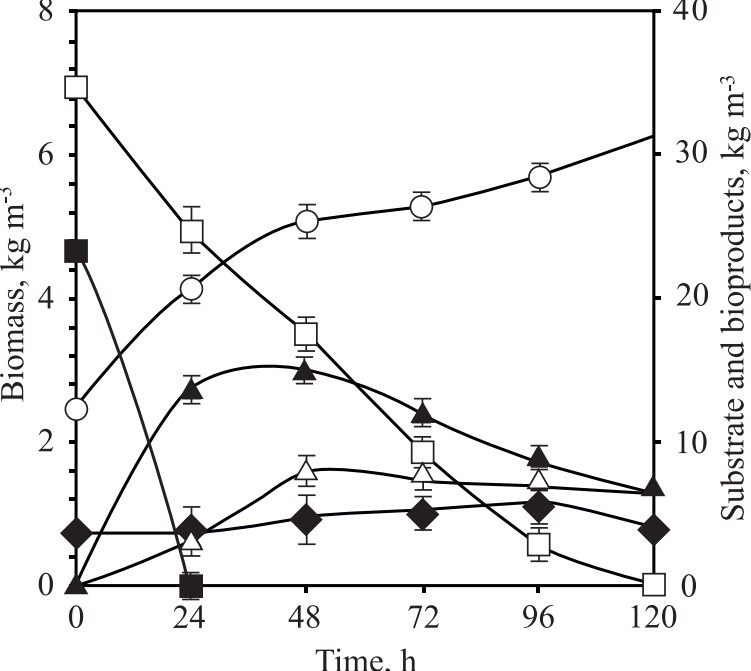

**Figure 4.** Fermentation experiment in Erlenmeyer flasks by *C. guilliermondii* ($x_0$ = 2.5 kg m$^{-3}$, pH = 5.5, T = 30 °C, $\omega$ = 0.4, agitation 200 rpm). Biomass (hollow circle), D-xylose (hollow square), D-glucose (filled square), ethanol (filled triangle), xylitol (hollow triangle) and acetic acid (filled rhomb).

After considering the most suitable values of the operation variables for Erlenmeyer flasks fermentations ($x_0$ = 2.5 kg m$^{-3}$, T = 30 °C, pH = 5.5, $\omega$ = 0.4 and 200 rpm), the same conditions were transferred to a bench bioreactor to evaluate the effect of scaling on *C. guilliermondii* fermentations but selecting the lowest of the possible stirring speed in the reactor (350 rpm), which implied (for an aeration of 0.4 vvm) a calculated $K_{La}$ of 17 h$^{-1}$ (value in line with [17]). The culture medium was inoculated and the fermentation process was monitored (Figure 5) so, under these experimental conditions, the yeast cells showed a behavior that resulted in a biomass volumetric productivity and biomass yield of 0.06 kg m$^{-3}$ h$^{-1}$ and 0.06 kg kg$^{-1}$, respectively (Table 4), which are both parameters in accordance with those obtained by García et al. [87] and Mateo et al. [88], using *C. tropicalis* and *D. hansenii*.

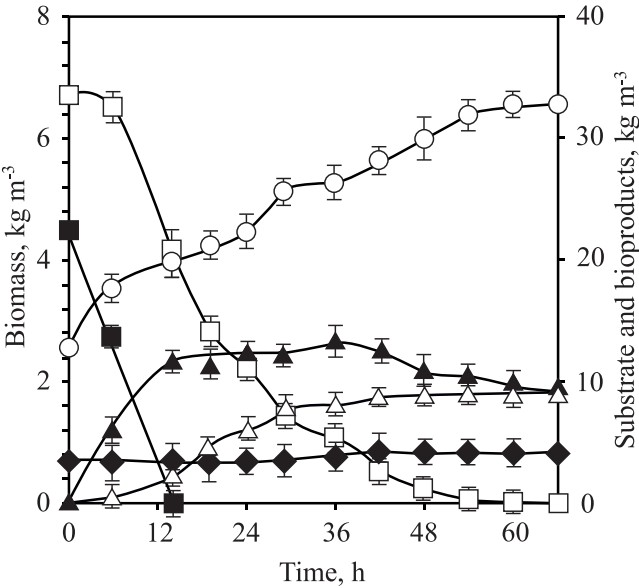

**Figure 5.** OTPB concentrated hemicellulosic hydrolysate fermentation in a bench bioreactor by *Candida guilliermondii*. Biomass (hollow circle), D-xylose (hollow square), D-glucose (filled square), ethanol (filled triangle), xylitol (hollow triangle) and acetic acid (filled rhomb).

**Table 4.** Fermentative parameters of the assays conducted on a bench bioreactor (OTPB-BB) and optimal conditions for Erlenmeyer flasks (OTPB-EF).

| Fermentative Parameters | OTPB-BB | OTPB-EF |
|---|---|---|
| $b$ (kg m$^{-3}$ h$^{-1}$) | 0.06 | |
| $Y_{x/s}^{G}$ (kg kg$^{-1}$) | 0.06 | $0.06 \pm 0.01$ |
| Initial D-glucose concentration (kg m$^{-3}$) | 33.44 | $34.88 \pm 0.15$ |
| Initial D-xylose concentration (kg m$^{-3}$) | 22.14 | $23.29 \pm 0.08$ |
| Maximum ethanol concentration (kg m$^{-3}$) | 13.32 | $14.91 \pm 0.12$ |
| Time to reach the maximum ethanol concentration (h) | 36 | 48 |
| $Y_{E/s}^{G}$ (kg kg$^{-1}$) | 0.28 | $0.38 \pm 0.01$ |
| $Q_E$ (kg m$^{-3}$ h$^{-1}$) | 0.84 | $0.57 \pm 0.01$ |
| Ethanol yield | 54.90 | $74.51 \pm 0.04$ |
| Maximum xylitol concentration (kg m$^{-3}$) | 9.13 | $8.11 \pm 0.11$ |
| Time to reach the maximum xylitol concentration (h) | 66 | 48 |
| $Y_{Xy/s}^{G}$ (kg kg$^{-1}$) | 0.37 | $0.46 \pm 0.02$ |
| $Q_{Xy}$ (kg m$^{-3}$ h$^{-1}$) | 0.26 | $0.17 \pm 0.01$ |
| Xylitol yield (% of theoretical value) | 40.66 | $50.55 \pm 0.09$ |

Initial monosaccharides concentrations in the experiment conducted on the bench bioreactor, OTPB-BB, were 33.44 kg m$^{-3}$ of D-xylose and 22.14 kg m$^{-3}$ of D-glucose

(Figure 5); simultaneous consumption of the two main sugars was observed, which was behavior verified by other authors [17,89] with no consumption of acetic acid, which has the dual inhibitor/substrate effect as described in the literature [28,67–70,90,91]. In addition, for the determination of the kinetic parameters, as shown in Table 4, the differential method of data treatment was used and, through an empirical adjustment, the specific rate of total substrate consumption was calculated based on the sum of D-glucose and D-xylose. In this sense, three stages of differentiated specific rate of substrate consumption were observed; during the first 5 h of the experiment, $q_s$ (kg kg$^{-1}$ h$^{-1}$) was maximized at 0.07; then, after 30 h, $q_s$ dropped to 0.01 and, finally, after 60 h, the specific rate was reduced to values less than 0.01 kg kg$^{-1}$ h$^{-1}$. Likewise, the ethanol formation from D-glucose and D-xylose ($Y_{E/s}^{G}$ = 0.28 kg kg$^{-1}$) began from the first stages of the experiment (Figure 5), while the first data on xylitol production were obtained later, after the depletion of D-glucose, reaching a xylitol yield value of 0.37 kg kg$^{-1}$. Based on the results obtained after scaling, the maximum ethanol and xylitol concentrations achieved when comparing both configurations for reaction systems (Erlenmeyer flasks and bioreactor) were quite similar. However, the ethanol and xylitol productivities were significantly higher in the bioreactor, but the bioproduct yields were slightly higher for Erlenmeyer fermentations of OTPB hydrolysate. The bioproduct yields presented promising results if they are compared to bibliography data. In this sense, Mateo et al. [81] performed fermentation assays of OTPB acid hydrolysates with *Candida tropicalis* reporting an ethanol yield (0.36 kg kg$^{-1}$, approximately) quite similar to that obtained in this investigation for trials conducted on Erlenmeyer flasks and a little higher value for the bench bioreactor one. Xylitol yields (0.22 kg kg$^{-1}$) were much lower than those shown in this work (Table 4), considering both Erlenmeyer flasks and the bioreactor. Likewise, the yields offered in the present research are similar, for some experimental conditions, or even much higher, than those offered by Pant et al. [92] when fermenting enzymatic hydrolysates of pretreated *Brassica juncea* with the yeast *Candida sojae*, displaying values for xylitol and ethanol yields in the range 0.08–0.52 kg kg$^{-1}$ and 0.04–0.28 kg kg$^{-1}$, respectively. It could be said that there is not enough information on the production of ethanol with *C. guilliermondii*, despite the fact that this yeast is a good producer of both ethanol and xylitol bioproducts, but research studies commonly focus the objectives on xylitol production [28,89,93,94].

## 4. Conclusions

OTPB hemicellulosic hydrolysate is a real and promising sugar source for xylitol and ethanol production as demostrated by the fermentation results attained by the non-traditional yeast *Candida guilliermondii* FTI 20037. On the other hand, it is necessary to continue with the process improvement, regarding volumetric productivities and yields, as well as the study of nutritional supplementation in order to minimize the extra-nutrients input, which would allow reducing the global process costs. In any case, the kinetic results obtained in a bench bioreactor allowed increasing the action scale, obtaining more real results, as one of the previous steps to enable mini-plant and industrial scaling.

**Author Contributions:** Conceptualization, I.C.R. and A.J.M.; methodology, J.G.P. and S.M.; software, A.J.M.; validation, S.S. and A.J.M.; formal analysis, I.C.R. and S.S.; investigation, J.G.P. and S.M.; resources, S.M.; data curation, J.G.P.; writing—original draft preparation, J.G.P.; writing—review and editing, S.M. and A.J.M.; visualization, S.M.; supervision, S.M. All authors have read and agreed to the published version of the manuscript.

**Funding:** This research received no external funding.

**Institutional Review Board Statement:** Not applicable.

**Informed Consent Statement:** Not applicable.

**Data Availability Statement:** The data generated during and/or analysed during the current study are available from the corresponding author on reasonable request.

**Acknowledgments:** The authors acknowledge the financial support of "Consejería de Economía, Innovación y Ciencia" from "Junta de Andalucía". Project No. AGR-6509, and the Department of Biotechnology on the Engineering College of Lorena, University of São Paulo (Brazil) for the research stage.

**Conflicts of Interest:** The authors declare no conflict of interest.

## Abbreviations

The following abbreviations are used in this manuscript:

| | |
|---|---|
| $A_{600}$ | Absorbance at 600 nm |
| $A_{760}$ | Absorbance at 760 nm |
| ASTM | American Society for Testing and Materials |
| $b$ | Biomass volumetric productivity |
| HMF | 5-hydroxymethylfurfural |
| HPLC | High-performance liquid chromatography |
| $K_{La}$ | Oxygen transfer volumetric coefficient |
| $\log R_0$ | Combined severity factor |
| NREL | National Renewable Energy Laboratory |
| OTPB | Olive tree-pruning biomass |
| OTPB-H | OTPB concentrated hydrolysate without addition of nutrients |
| OTPB-LH | OTPB concentrated hydrolysate with addition of nutrients proposed by [36] at 50% |
| OTPB-RH | OTPB concentrated hydrolysate with addition of nutrients proposed by [37] |
| $Q_E$ | Ethanol volumetric productivity |
| $q_s$ | Specific rate of substrate consumption |
| $Q_{Xy}$ | Xylitol volumetric productivity |
| SM-L | Substrate constituted by the addition of nutrients proposed by [36] at 50% |
| SM-R | Substrate constituted by the addition of nutrients proposed by [37] |
| $TFC$ | Total phenolic compounds concentration |
| $Y^G_{x/s}$ | Biomass global yield |
| $Y^G_{E/s}$ | Ethanol global yield |
| $Y^G_{Xy/s}$ | Xylitol global yield |
| $\omega$ | Aeration |

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
