# Peer review of "Bioconversion Study of Olive Tree Biomass Hemicellulosic Hydrolysates by Candida guilliermondii at Different Scales for Ethanol and Xylitol Production"

_fermentation, doi:10.3390/fermentation9060553_

Round 1

Reviewer 1 Report

Manuscript ID: fermentation-2403329

Title: Bioconversion study of olive tree biomass hemicellulosic hydrolysates by Candida guilliermondii at different scales for ethanol and xylitol production

The authors studied and optimised the fermentation of the Olive tree pruning biomass hydrolysate in Erlenmeyer flasks and in a bench bioreactor by the yeast Candida guilliermondii FTI 20037 to produce ethanol and xylitol. The xylose-rich hydrolysate was obtained by using H2SO4 as catalyst for the chemical hydrolysis of the hemicellulose fraction of the starting biomass. Firstly, the medium composition was optimised in terms of nutrients composition and concentration by comparing synthetic media and the real OTPB hydrolysate. Then the effect of temperature, initial inoculum concentration and oxygenation on the fermentation performance was investigated. In the end, the optimal values of the operating variables for Erlenmeyer flask fermentations were transferred to a bench bioreactor to evaluate the effect of scaling on C. guilliermondii fermentation performance.

The paper is well-developed and clearly readable, the results are quite interesting, and the topic fits well into the journal Fermentation. However, the manuscript needs a major revision before it can be accepted for publication. As follows:

Major comments:

1.      The authors should better emphasise the innovative aspects of the present work in the manuscript.

2.      In the Materials and Methods Section, the authors should add the description of experimental conditions used for the chemical hydrolysis of the biomass in order to obtain the OTPB hydrolysate.

3.      In the Materials and Methods Section, the authors should add the important information related to: i) yeast strain source and pre-cultivation; ii) batch-mode fermentation conditions in Erlenmeyer flasks and bench bioreactor. Were the fermentations performed in triplicate?

4.      In the caption of Figure 1 the authors wrote “Sugars and inhibitors yield..” but the values reported in the figure are concentrations (kg/m3) in agreement with the information reported in lines 228-233. Please revise Figure 1 and the text (e.g. line 182) accordingly.

5.      The authors should report and discuss the glucose and xylose yields (expressed as mol% with respect to the glucan and xylan content in the starting biomass) obtained after the H2SO4-catalysed hydrolysis of OTPB. Is this approach innovative for this biomass? Are the obtained sugar yields higher than the values reported in the literature for the same biomass with the same or different catalytic approaches for the production of fermentable sugars?

6.      The comparison of the results achieved in the present study with respect to those reported in the literature for the same yeast (in terms of sugars consumption rate, biomass growth rate, ethanol and xylitol yield, production and productivity) should be better discussed by the authors.

7.      The authors should add the statistical analysis of their fermentation results (e.g. ANOVA analysis) in order to demonstrate the statistically significant effect of the different variables that were tested in the present study (e.g. effect of nutrients, pH, inoculum size, scale-up).

8.      What is the fate of the cellulose-rich solid residue obtained after the acid hydrolysis of the hemicellulose fraction? Could this cellulose-rich residue be valorised? The authors should point out these aspects from the perspective of the complete valorisation of the selected lignocellulosic biomass.

Minor comments:

1.      Lines 34-35: the authors should also mention the detoxification by vacuum evaporation as reported by Di Fidio et al. (2019 https://doi.org/10.4014/jmb.1808.08015, 2021 https://doi.org/10.1016/j.biortech.2020.124635) for the detoxification of the xylose-rich hydrolysates obtained from lignocellulosic biomasses.

2.      Lines 108-132: in the Section “2.2. General Analytical methods” the authors should add all the equations used for the determination of parameters reported and discussed in the manuscript, including the equations for the calculation of sugars yields (mol%) after the hydrolysis step.

3.      Lines 136-148: the authors should also mention giant reed as typical and strategic hemicellulose-rich biomass for biorefinery schemes as reported by De Bari et al. (https://doi.org/10.1016/j.apenergy.2012.05.051) and by Di Fidio et al. (https://doi.org/10.1016/j.biortech.2020.124635).

4.      Figure 1 should be reported in the Section “3.2 OTPB hemicellulosic hydrolysate” and not before.

5.      The authors should add the error bars in Figures 2, 4 and 5.

Author Response

We have responded to the comments in the attached file.

Reviewer 2 Report

The manuscript need to be restructured. Concerns are mainly regarding description of the research plan, materials, methods and equipment used to reach the obtained results. In this form of the manuscript is very difficult for the readers to follow the path of the research, to analyse the methods applied and equipment used in this research and to compare those tools with own approaches or to the state of the art in the field. Publishing the article in this form would not be attractive for readers and citing. 

 Moderate editing of English language required

Author Response

We have improved and restructured the paper, including the description of the research plan and mainly the Materials and methods section. Thus, specific subsections have been introduced for the description of acid hydrolysis, yeast strain source and pre-cultivation, and fermentation conditions. In addition, the different equations used to calculate the different parameters have been provided. Finally, comparisons of the results obtained by other authors have been added. We hope that all these improvements allow readers to follow the path of the research and represent a significant improvement that allows its publication.

Reviewer 3 Report

The manuscript examines the bioconversion of olive tree prunings (without leaves) into ethanol and xylitol using a selected Candida strain. Namely, the authors have searched for the most optimal process conditions in Erlenmayer flasks and then determined its kinetics under these conditions in a lab bioreactor. However, some key research points have been missed in order for this manuscript to be acceptable for publication.

The following issues need to be addressed before the manuscript can be seriously reviewed:

1. The Materials and Methods section is missing the hydrolysis, concentration and fermentation sections, in order to have all the details regarding the performed experiments. Some of this data is mentioned in the Results and Discussion section but it is not clear and it is uncertain what has been done. Also, it would be useful to present all the equations or methods of how the parameters were calculated (severity factor, yields, productivity, etc.)

2. Figure 1 could also present the concentration of the components in the raw and concentrated hydrolysate. This is of more use than the yield itself.

3. If nutrients were added to the hydrolysate in 50 and 100% for forming the fermentation media, why wasn't 0% also one of the options?

4. The process of obtaining the optimal process parameters is quite confusing. How did the authors choose five different scenarios from the 27 possible when examining the effect of three parameters on three levels (temperature, inoculum concentration and aeration at high, medium and low values)? And after that, they examine the other two (pH value and agitation speed) separately. And finally, how was the optimum defined/chosen? If there is a methodology for this please describe it in the Materials and Methods section.

5. Figure 4 shows that the ethanol and xylitol concentrations (main products and aim of this study) reach their maximal concentrations in the fermentation broth at 48 h, after that they are either constant (xylitol) or decline (ethanol), meaning that fermentation time has also a major role on influencing the process?

6. The optimal agitation speed in the Erlenmayer experiment is 200 rpm in the Results and Discussion section, but 250 in the abstract.

7. If the bioreactor has the lowest possible agitation speed of 350 rpm, why were even lower values examined for this parameter in the Erlemayer experiments?

8. Check the entire manuscript for typing errors like Line 6 delete bracket after 200 rpm, missing space between words in Lines 123 and 129, missing bracket in Line 156, etc. There is a typo in the caption of Figure 3: pH = 5.5 kgm-3.

9. Some sentences need to be rephrased and checked because their meaning is not clear (Lines 12, 24-32, 183, Figure 2 caption, etc.)

10. Third paragraph of the Introduction section should be like the rest and not like the final paragraph, i.e. containing only a literature overview and not what has been done and what will be presented in this manuscript. That is always left for the final paragraph.

Author Response

(The authors gave the same response as above.)

Reviewer 4 Report

1.       Olive tree pruned biomass analysis, pretreatment and fermentation studies have been extensively published.  New to this study is the use of a specific strain of candida and bench-level scaling-up.  Given this, the authors should compare their work/results with similar studies and show how their work is novel, especially as related to Ethanol and Xylitol production.

2.       The title of the paper should be concise statement.

3.       The paper has made no efforts to emphasize on the novel of the study.

4.       The biggest challenge with this manuscript is how poorly it is written?  It needs extensive English editing before one can clearly understand what the authors are trying to convey.  The sentences are run-on to paragraphs at time mixed with present to past and to future sentences.  This must be corrected!

Very poorly written paper.  Needs extensive editing.

Author Response

(The authors gave the same response as above.)
